# Shear Wave Elastography for measuring the elastic properties of the Psoas Major muscle: An intra- and inter-examiner reliability study

Gabriel Rabanal-Rodríguez[1,2], César Fernández-de-las-Peñas[3,4], Javier Álvarez-González[5,6], Alberto Roldán-Ruiz[5], Jorge Buffet-García[5*], Juan Antonio Valera-Calero[1,2]

1 Department of Radiology, Rehabilitation and Physiotherapy, Universidad Complutense de Madrid, Madrid, Spain, 2 Grupo InPhysio, Instituto de Investigación Sanitaria del Hospital Clínico San Carlos (IdISSC), Madrid, Spain, 3 Department of Physical Therapy, Occupational Therapy, Rehabilitation and Physical Medicine, Universidad Rey Juan Carlos, Alcorcón, Spain, 4 Cátedra Institucional en Docencia, Clínica e Investigación en Fisioterapia: Terapia Manual, Punción Seca y Ejercicio Terapéutico, Universidad Rey Juan Carlos, Alcorcón, Spain, 5 Faculty of Health Sciences, Universidad Francisco de Vitoria, Pozuelo de Alarcón, Spain, 6 Department of Radiology, Hospital General Universitario Gregorio Marañón, Madrid, Spain

* j.buffet.prof@ufv.es

## Abstract

### Background

The psoas major (PM) plays an important role in population with low back pain (LBP). Its evaluation considers clinical history, which can be confused with other lumbar and lower limb conditions, physical examination, with an inconsistent palpation, and imaging tests that provide inconclusive parameters in various studies. Therefore, developing reliable assessment procedures to evaluate PM elastic properties is necessary to improve diagnosis and follow-up. This study aimed to evaluate the intra and inter-examiner reliability of shear wave elastography (SWE) for calculating the PM stiffness in patients with LBP.

### Design

Observational study.

### Methods

Longitudinal views of the of the PM muscle using SWE were acquired bilaterally in 52 volunteers experiencing moderate LBP and disability. All measurements were performed twice, involving the assessment of shear wave speed and Young's modulus as indicators of stiffness, with data collected by an experienced examiner and a novice examiner.

**Data availability statement:** All relevant data are within the paper and its Supporting information files.

**Funding:** This study was financially supported by Banco Santander (https://www.bancosantander.es) in the form of a researcher training scholarship award (CT25/24) received by GR-R. The funder had no role in study design, data collection and analysis, decision to publish, or preparation of the manuscript. No additional external funding was received for this study.

**Competing interests:** The authors have read the journal's policy and confirm there are no patents, products in development or marketed products associated with this research to declare. The authors also confirm adherence to PLOS ONE policies on sharing data and materials.

## Results

Test-retest reliability showed strong consistency regardless of the examiner's level of experience, with intraclass correlation coefficients (ICCs) exceeding 0.9 for both metrics. However, experienced examiners achieved smaller minimal detectable changes. Inter-examiner reliability was comparatively lower, with ICCs ranging from 0.854 to 0.925, and notable differences in mean values between examiners were observed ($p < 0.01$).

## Conclusion

Excellent test-retest reliability was observed for the measurement of PM muscle stiffness in chronic LBP patients using SWE by both experienced and novice evaluators, although statistically significant differences were found between the two trials for the novice examiner. Inter-examiner reliability was lower, highlighting variability between assessors. To minimize errors and improve consistency and accuracy, if multiple examiners are involved, averaging measurements is recommended.

## Introduction

The psoas major (PM) is a long muscle that originates from the transverse processes, vertebral bodies, and intervertebral discs of T12 to L5 and inserts into the lesser trochanter of the femur. Biomechanically, the PM exerts compression and shear forces on the lumbar spine, tending to produce extension moments in the upper lumbar segments and flexion moments in the lower segments. Despite its influence on lumbar spine movements, its primary function is hip flexion, with a secondary role in lumbar stabilization through force transmission during trunk and hip flexion activities [1].

From a clinical perspective, the PM muscle plays a key role in the pathogenesis of painful musculoskeletal conditions affecting the lumbar spine and the lower limbs (e.g., groin injuries, low back pain (LBP), pelvic pain, hip snapping and femoroacetabular impingement) [1] as described in numerous reports [2–7].

Regarding non-specific LBP, it has been established that increased PM stiffness due to continued local contraction could be a relevant etiologic factor, suggesting that tools for its evaluation, like elastography, can provide significant quantitative information for the diagnosis and assessment of these patients [8]. In this context, other authors have also noted overactivation of the PM in patients with chronic LBP [7].

The evaluation of the PM is based on the medical history, clinical examination and complementary imaging tests. Isolated assessment based on clinical history is complex and not recommended, since it may present compatible symptoms with impairments or injuries of other structures of the lumbopelvic region. In the clinical setting, the evaluation can be complemented by palpation (with evidence advising that soft tissue palpation is inconsistent in patients with LBP [9]) or by performing muscular resistance tests (with isometric contractions) or orthopedic tests including

the Thomas or FADIR maneuvers [1]. As for imaging tests, the gold standard is considered to be Magnetic Resonance Imaging, although some studies have also used Computerized Tomography [5]. However, in recent years, ultrasound imaging (US) has begun to be studied as a possible alternative for the evaluation of the muscular characteristics of the PM [10–12].

This tool, in particular through sonoelastography, allows to assess tissue elasticity, functioning as an adjunct to B-mode US and offering significant advantages with reduced cost, examination time and complexity. Shear wave elastography (SWE), which employs mechanical shear waves generated by the compressive acoustic waves used during B-mode imaging to measure the velocity of shear wave propagation (which correlates directly with tissue stiffness), despite competing with other elastography techniques for the assessment of musculoskeletal tissues [13], is regarded as the most reliable approach. Evaluating muscle stiffness could offer clinically significant insights, potentially identifying early disease stages when morphological changes are not yet visible on grayscale imaging, making it more informative than parameters such as size, histology, or shape descriptors.

Taking into account the available evidence on changes in the muscular characteristics of the PM in relation to LBP and disc pathology [6,14–22], the uncertain physical examination with an inconclusive palpation and given the scarce published literature on the usefulness and reliability of US assessment for the determination of its properties, only including the study by Zhou et al. [23] in healthy subjects, whose results may not be extrapolated to individuals with LBP regarding their particular characteristics, the development of a study to establish valid and reliable methods for the assessment of the elastic properties of the PM is rationally justified.

This information could enable researchers to continue with a line of knowledge on the association between SWE-obtained parameters and LBP-related variables as suggested on clinical practice guidelines [24], as well as deepening into SWE precision for the discrimination between LBP and non-LBP populations.

Thereby, the primary objective of this study is to evaluate the intra and inter-examiner reliability of a SWE technique for assessing the stiffness of the PM muscle in a cohort of patients experiencing LBP.

## Methods

### Study design

Between 29th February and 31st May 2024, a longitudinal observational study was carried out to evaluate the intra and inter-examiner reliability estimates of a SWE procedure. The study was conducted in a physiotherapy laboratory within the Faculty of Health Sciences of the Francisco de Vitoria University. To ensure the quality of the report, the guidelines outlined in the Reporting Reliability and Agreement Studies (GRRAS) [25] and Enhancing the QUAlity and Transparency Of health Research (EQUATOR) frameworks [26] were followed. Furthermore, participants' rights were safeguarded in compliance with the Declaration of Helsinki, and the study protocol was reviewed and approved by the Ethics Committee of the Francisco de Vitoria University (15/2024) prior to data collection.

### Participants

A sample of patients with LBP was recruited through local announcements posted at the campus of the Francisco de Vitoria University. The inclusion criteria required participants to be between 18 and 65 years old and to have experienced at least one clinically significant episode of LBP within the past year, without neurological signs, along with mild-to-moderate pain intensity (minimum 3 out of 10 on the Visual Analogue Scale [27]) and associated disability (at least 12 out of 100 on the Oswestry Disability Index [28]) at the time of data collection.

Individuals were excluded if they were undergoing pharmacological treatment that could influence muscle tone, had a history of spinal surgeries, neuropathies (e.g., radiculopathies or myelopathies), serious medical conditions (e.g., tumors, fractures, neurological disorders, or systemic diseases), clinically relevant asymmetries, or widespread musculoskeletal conditions such as fibromyalgia.

Participants who met these criteria were asked to read, understand, and sign a written informed consent form before being scheduled for data collection.

## Sample size calculation

The minimum sample size required for this study was calculated using the methodology outlined by Walter et al. [29] for reliability studies involving intraclass correlation coefficients (ICCs). Since no previous studies had evaluated intra or inter-examiner reliability estimates in clinical populations, data from asymptomatic subjects were used as a reference. Zhou et al. [23] reported test-retest reliability ICCs ranging from 0.89 to 0.92 in a sample of 52 subjects.

Based on these references, the following parameters were applied: the minimum acceptable ICC ($\rho 0$) was set at 0.7 (the lower limit of "good reliability" as defined in the literature [30]), the expected ICC ($\rho 1$) at 0.92, the significance level ($\alpha$) at 0.05, the statistical power ($1 - \beta$) at 0.95, with 2 raters (k) and an anticipated dropout rate of 10% due to the longitudinal nature of the study. Consequently, a total of 29 participants were required to ensure adequate statistical power.

## Examiners

The study involved two examiners: one experienced examiner with over 10 years of expertise in musculoskeletal US and clinical management of musculoskeletal conditions, and one novice examiner with less than 1 year of experience in both areas who had completed 20 hours of US training.

To enhance methodological rigor (as assessing the left and right side from the same participant cannot be fully considered as independent datapoints), imaging acquisition procedures were randomized for both the order in which volunteers participated and the initially examined side. For each measurement, the entire positioning process of the participant and the probe was restarted to minimize the potential influence of prior alignment or muscle relaxation. The study was conducted in two separate sessions: the first in the morning, between 9:00 and 11:00, and the second in the afternoon, from 13:00–15:00. A strict isolation protocol was followed, with the two examiners working on alternating days to eliminate any possibility of communication or agreement between them.

Participants were asked to attend twice a day for two consecutive days (24-hour interval). The first two visits were in day one (in the morning with examiner A and in the afternoon with examiner B) and the second two visits were in day two (in the morning with examiner B and in the afternoon with examiner A). A single image was obtained in each session. Subsequently, a third investigator coded all the images. Finally, each examiner measured their own images in a randomized order, ensuring blinding to both the participant's identity and the evaluated side.

## US acquisition protocol

The US device used for image acquisition was a Canon Aplio A, equipped with a convex transducer 8C1 (Canon Medical Corp, 1385 Shimoishigami, Otawara, Tochigi 324–8550, Japan). Standard console settings were applied for all acquisitions (Frequency: 5 MHz, Gain: 80 dB, Dynamic Range: 60, Depth: 12 cm).

The imaging acquisition protocol adhered to the guidelines established by Zhou et al. [23]. Participants were positioned in a lateral decubitus posture to ensure spinal and lower limb neutrality. A wedge-shaped cushion was placed posterior to the upper thoracic region to maintain the torso perpendicular to the examination table. Additionally, a square cushion was inserted between the thighs to maintain neutral hip alignment, and a towel was used when necessary to support a neutral lumbar spine position. Participants were instructed to relax their muscles during the procedure to prevent morphological bias caused by muscle contractions [31].

Acoustic coupling gel was applied to the transducer, which was initially positioned superior to the iliac crest along the mid-axillary line. The cranial end of the transducer was then tilted posteriorly by approximately 20° to center the L4

vertebra in the imaging field using B-mode US (Fig 1A). At this location, the PM muscle was identified as the structure overlying the vertebral bodies (Fig 1B).

To capture images of the PM, the transducer's orientation was adjusted to align parallel to the muscle fibers' long axis, ensuring the central portion of the transducer was perpendicular to the muscle fibers. Once aligned, the SWE mode was activated, positioning the region of interest in the center of the muscle and covering at least 50% of its area to capture the image, adapting the dimensions to each participant depending on their characteristics. Finally, images were measured by outlining a freely drawn quantification box, ensuring it did not overlap the muscle edges and excluded the muscle fasciae (Fig 1C). All SWE images were acquired using the built-in software of the Canon Aplio A system, which automatically applies standard post-processing algorithms to optimize image quality and measurement accuracy. These include real-time smoothing filters to reduce noise and enhance the visibility of shear wave propagation, as well as temporal averaging to stabilize the color-coded elastograms. The system also applies quality maps to guide the placement of the region of interest, ensuring that measurements are taken only from areas with sufficient signal stability. No manual or external image processing was performed after acquisition.

After contouring the region of interest, the US device automatically provides the shear wave speed and automatically calculates the Young's modulus. The relationship between these two parameters is based on the physical equation $E = 3\rho \cdot SWS^2$ where E is Young's modulus (in kilopascals), $\rho$ is the tissue density (usually assumed to be $\sim 1000\,kg/m^3$ for soft tissue), and SWS is the shear wave speed (in meters per second). This formula assumes the tissue behaves as a linear, isotropic, and purely elastic medium. Although biological tissues are more complex, this approximation allows the ultrasound system to express stiffness in both m/s (direct measure of SWS) and kPa (converted Young's modulus), depending on clinical preference or research requirements.

The InBody 770 bioimpedance device (Biospace, Urbandale, IA, USA) was used to assess body composition parameters, collecting values for water volume, body weight, body mass index (BMI) and muscle and body fat mass.

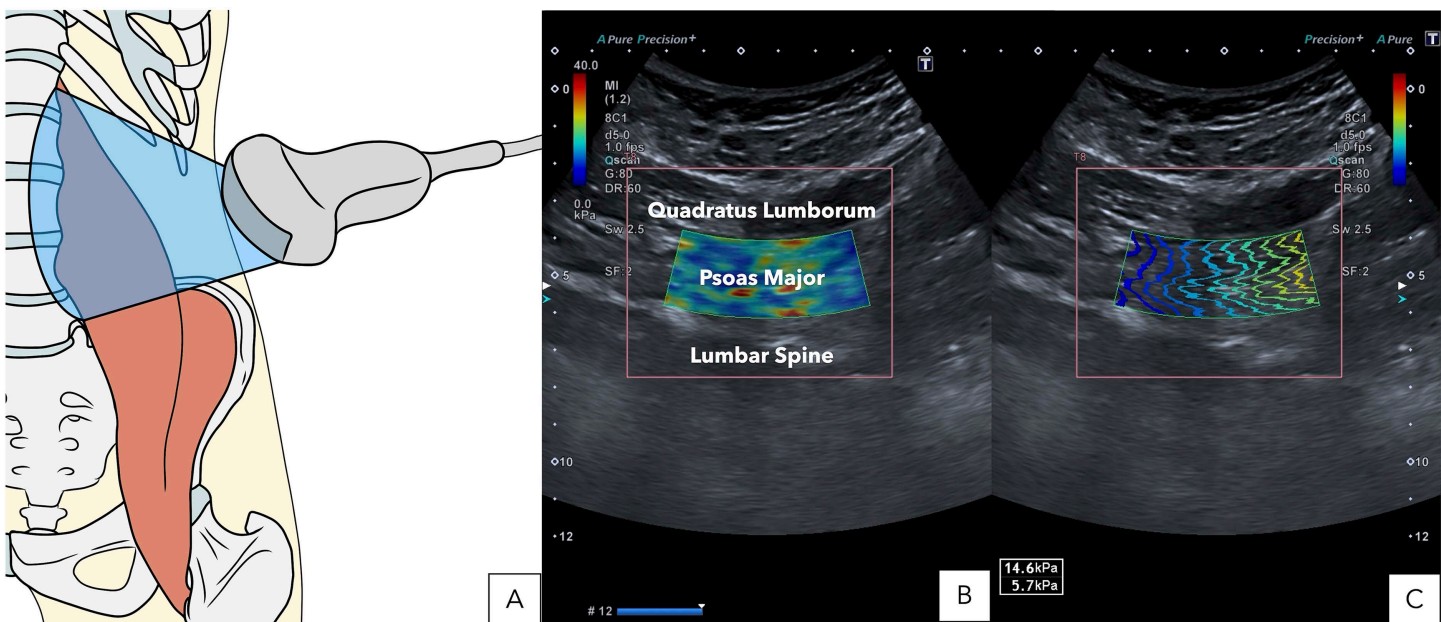

**Fig 1. Transducer positioning and an illustrative example of shear-wave elastography image.** (A) Transducer positioning; (B) Shear wave imaging identifying the most relevant musculoskeletal structures: quadratus lumborum, psoas major and lumbar spine; (C) Quantitative measurement of shear wave imaging: Young's modulus (kPa) and shear wave speed (m/s).

## Statistical analysis

All data processing and analysis were performed using the Statistical Package for the Social Sciences (SPSS) v.29.1.1 (Armonk, NY, USA) for Mac OS. All tests were two-tailed, with a significance threshold set at $p < 0.05$. The distribution of continuous variables was initially evaluated using histograms and Shapiro-Wilk tests.

The demographic and clinical characteristics of the sample were described using descriptive statistics. For categorical variables, frequencies and percentages were reported. Continuous variables were described using measures of central tendency (mean or median) and dispersion (standard deviation or interquartile range), depending on whether the data followed a normal distribution. Gender differences were analyzed using Student's T-tests for independent samples, reporting the mean difference, 95% confidence intervals, and p-values.

For the reliability analyses, the following metrics were calculated: 1) The mean and standard deviation for each US metric obtained by both operators; 2) The measurement disagreement between examiners, computed as the difference between trials for intra-examiner reliability and as the difference between examiners for single measures and the average of two attempts for inter-examiner reliability, along with the absolute error to avoid underestimating errors; 3) ICCs (ICC3,1 for intra-examiner reliability and ICC3,2 for inter-examiner reliability) using a two-way mixed-effects model for consistency; 4) The standard error of measurement (SEM), calculated as the standard deviation of the mean multiplied by the square root of 1 minus the ICC; 5) The minimal detectable change (MDC), determined as 1.96 times the square root of 2 times the SEM [30]; and 6) the coefficient of variation, calculated to express the relative variability of the measurement error with respect to the average value of the parameter assessed. It was calculated by dividing the absolute error by the mean of the corresponding variable. The result was then multiplied by 100 to express the coefficient of variation as a percentage.

Classification tresholds for ICCs were established following the general guidelines proposed by Koo et al. [30], assigning poor reliability for values under 0.5, moderate reliability for values between 0.5 and 0.75, good reliability for values between 0.75 and 0.9 and excellent reliability for values exceeding 0.9.

Finally, Bland-Altman plots were generated for both shear wave speed and Young's modulus across examiners. For each parameter, the difference between Trial 1 and Trial 2 values was plotted against their mean. The plots include the mean bias (solid red line) and the 95% limits of agreement (mean difference ±1.96×standard deviation of the differences; dashed lines). This approach allows for visual inspection of the consistency and spread of repeated measurements. Additionally, to assess the presence of systematic error, we examined whether the mean difference significantly deviated from zero using paired-sample t-tests. Proportional bias was evaluated by performing linear regression analyses between the differences and the means of the two trials; a significant regression slope would indicate proportional error. These analyses were conducted separately for novice and experienced examiners, and for each measurement parameter, to assess the influence of examiner expertise on measurement agreement. The database is available in Supporting information.

## Results

Throughout the recruitment phase, 52 individuals indicated a willingness to participate in the study. All prospective participants met the inclusion criteria, and therefore none were excluded. Every image acquired during data collection was reviewed in terms of content and quality, accepted and subsequently analyzed, ensuring that no data were lost. The final sample comprised 35 women and 17 men, and 103 images of the PM muscle were captured from both sides. Given that each examiner acquired two images per muscle, a total of 416 SWE images were acquired and 412 images (n = 4 were excluded due to a data loss) were analyzed.

Table 1 provides an overview of the demographic and clinical profiles of the participants, including gender-based comparisons. Statistically significant differences in demographic variables were found between male and female participants: males were notably younger (p < 0.001), heavier (p < 0.001), and taller (p < 0.001). Differences were also observed in body composition; males exhibited a higher BMI (p = 0.009) and greater water volume (p < 0.001). Nonetheless, despite the BMI discrepancy, there was no significant difference in body fat percentage between sexes (p = 0.271). Clinically, both genders

**Table 1. Demographic and clinical characteristics of the sample analyzed.**

| Variables | Subjects with Low Back Pain (n=52) | | Difference (95% CI) |
| --- | --- | --- | --- |
| | Females (n=35) | Males (n=17) | |
| Demographics | | | |
| Age, years | 32.5±13.4 | 24.6±7.7 | 7.9 (0.6;15.3) p<0.001 |
| Weight, kg | 69.9±13.6 | 92.8±13.8 | 22.9 (14.3;31.4) p<0.001 |
| Height, m | 1.65±0.04 | 1.76±0.08 | 0.11 (0.08;0.15) p<0.001 |
| BMI, kg/m² | 25.7±5.1 | 29.9±4.9 | 4.2 (1.1;7.3) p=0.009 |
| Water volume, L | 32.3±3.17 | 48.5±8.8 | 16.1 (10.6;21.6) p<0.001 |
| Body fat, % | 33.0±10.5 | 25.4±12.2 | 7.6 (−6.5;21.7) p=0.271 |
| Clinical Characteristics | | | |
| VAS, 0–10 | 4.9±1.7 | 5.2±1.9 | 0.3 (−0.8;1.4); p=0.584 |
| ODI, 0–100 | 24.8±9.2 | 25.3±9.0 | 0.5 (−0.2;1.3) p=0.209 |

n: Number; CI: Confidence interval; BMI: Body Mass Index; VAS: Visual Analogue Scale; ODI: Oswestry Disability Index.

reported comparable levels of pain intensity, classified as moderate (p=0.584), and similar degrees of disability (moderate disability, p=0.209).

Reliability estimates for repeated measures of PM stiffness are shown in Table 2. For the experienced examiner, no significant differences emerged between the first and second trials in measurements of shear wave speed or Young's modulus (both p>0.05). Conversely, the novice examiner demonstrated significant trial-to-trial differences in both parameters (shear wave speed: p=0.041; Young's modulus: p=0.048). Despite these discrepancies, both examiners achieved excellent intra-class correlation coefficients (ICCs>0.9), although the experienced examiner exhibited superior

**Table 2. Test-retest reliability estimates to determine psoas major stiffness.**

| | Experienced Examiner | | | Novice Examiner | | |
| --- | --- | --- | --- | --- | --- | --- |
| | Trial 1 (n=103) | Trial 2 (n=103) | | Trial 1 (n=103) | Trial 2 (n=103) | |
| Shear Wave Speed (m/s) | | | | | | |
| Mean | 2.14±0.32 | 2.15±0.30 | | 2.03±0.27 | 2.09±0.32 | |
| Error | −0.01±0.10 | | | −0.06±0.15 | | |
| Absolute Error | 0.08±0.06 | | p>0.05 | 0.13±0.09 | | p=0.041 |
| ICC | 0.971 (0.957;0.980) | | | 0.930 (0.897;0.953) | | |
| SEM | 0.05 | | | 0.15 | | |
| MDC | 0.15 | | | 0.43 | | |
| CV (%) | 3.7 | | | 6.3 | | |
| Young's Modulus (kPa) | | | | | | |
| Mean | 15.79±4.91 | 15.92±4.54 | | 15.02±4.59 | 16.03±5.67 | |
| Error | −0.12±1.40 | | | −1.01±3.05 | | |
| Absolute Error | 1.08±0.89 | | p>0.05 | 2.62±1.85 | | p=0.048 |
| ICC | 0.977 (0.966;0.985) | | | 0.904 (0.858;0.935) | | |
| SEM | 0.74 | | | 2.61 | | |
| MDC | 2.06 | | | 7.24 | | |
| CV (%) | 6.8 | | | 16.9 | | |

n: Number; CV: Coefficient of Variation; ICC: Intraclass Correlation Coefficient; SEM: Standard Error of Measurement; MDC: Minimal Detectable Change.

consistency across trials (shear wave speed: ICC = 0.971 vs. ICC = 0.930; Young's modulus: ICC = 0.977 vs. ICC = 0.904). Furthermore, the experienced examiner showed a lower minimal detectable change (MDC), suggesting greater sensitivity to true changes beyond measurement error (shear wave speed: MDC = 0.15 vs. 0.43; Young's modulus: MDC = 2.06 vs. 7.24). Bland-Altman plots (Fig 2) visually represent these reliability findings, plotting the mean of both trials on the X-axis and their differences on the Y-axis.

Table 3 presents inter-examiner reliability metrics. Analysis of mean differences revealed that most of the measurements taken by the two examiners differed significantly (shear wave speed: both single and averaged measurements, p = 0.001; Young's modulus: single measurement, p = 0.01). However, no statistically significant difference was found when comparing the average of two Young's modulus measurements (p > 0.05). Reflecting these outcomes, inter-examiner ICC values were generally lower than intra-examiner values. Single measurements showed good reliability between examiners (shear wave speed: ICC = 0.854; Young's modulus: ICC = 0.858). When the average of two measurements was used, reliability improved to good-to-excellent levels (shear wave speed: ICC = 0.857; Young's modulus: ICC = 0.925). Correspondingly, MDC values were reduced when using averaged measurements, aligning with the improved ICCs.

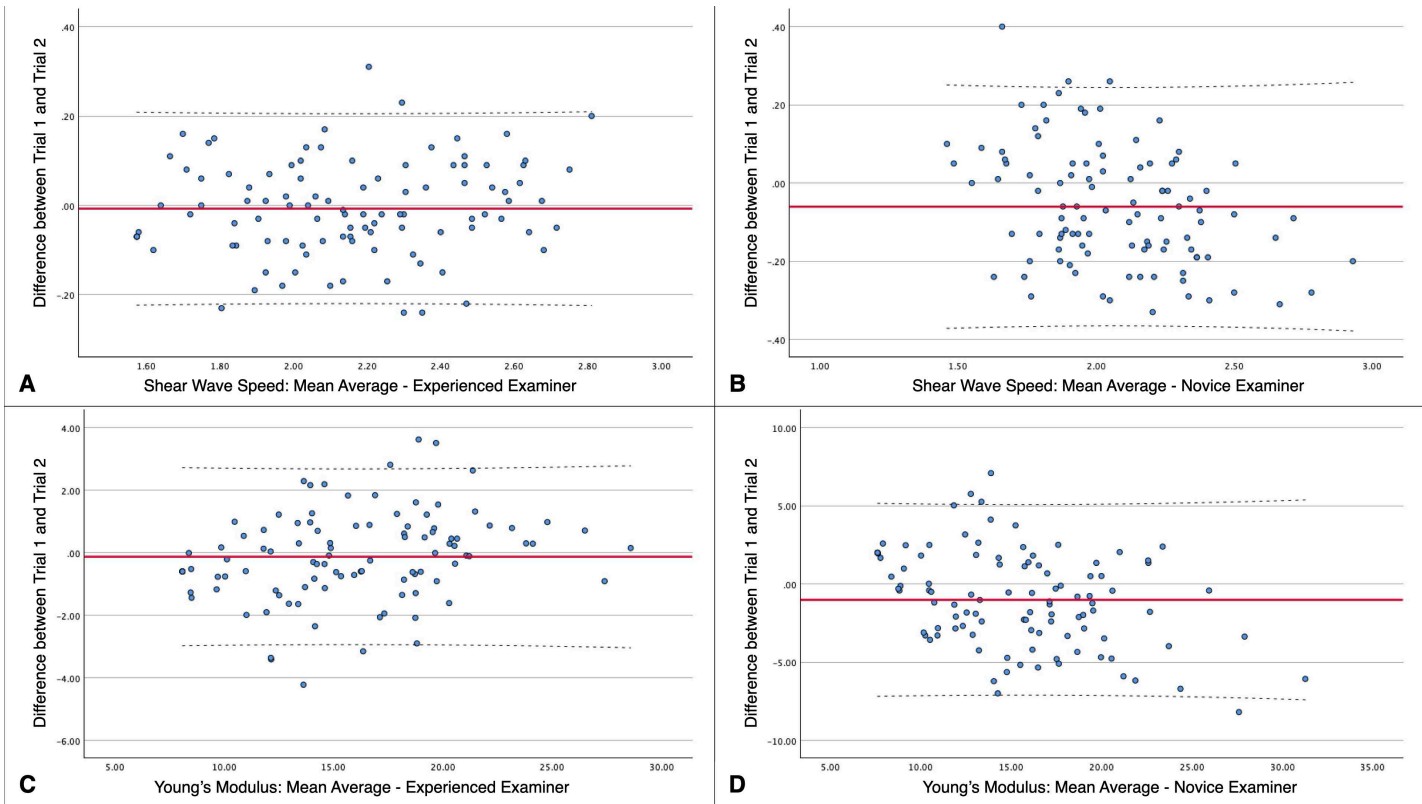

**Fig 2. Bland-Altman plots comparing the difference between Trial 1 and Trial 2 measurements across different examiners and parameters.** **(A)** Shear wave speed: mean average for an experienced examiner, **(B)** Shear wave speed: mean average for a novice examiner, **C)** Young's Modulus: mean average for an experienced examiner, **(D)** Young's Modulus: mean average for a novice examiner. The solid red line represents the mean difference (bias) between trials, while the dashed lines indicate the 95% limits of agreement (mean difference ± 1.96 × standard deviation of the differences). Note: the dashed lines may appear slightly curved due to the smoothing method used by the plotting software, but they reflect constant limits of agreement across the x-axis.

**Table 3. Inter-examiner reliability analysis: Single and mean average-measures scores to determine psoas major stiffness.**

| | 1 measurement (n = 103 images per examiner) | Mean average of 2 measurements (n = 206 images per examiner) |
|---|---|---|
| Shear Wave Speed (m/s) | | |
| Mean | 2.09 ± 0.20 | 2.10 ± 0.20 |
| Error | 0.11 ± 0.43 | 0.02 ± 0.24 |
| Absolute Error | 0.37 ± 0.25 p = 0.001 | 0.19 ± 0.13 p = 0.001 |
| ICC | 0.854 (0.785;0.901) | 0.857 (0.788;0.903) |
| SEM | 0.07 | 0.07 |
| MDC | 0.21 | 0.21 |
| CV (%) | 17.7 | 9.0 |
| Young's Modulus (kPa) | | |
| Mean | 15.40 ± 3.14 | 15.69 ± 3.23 |
| Error | 0.76 ± 7.13 | −0.12 ± 3.86 |
| Absolute Error | 5.96 ± 3.94 p = 0.01 | 3.15 ± 2.21 p > 0.05 |
| ICC | 0.858 (0.790;0.904) | 0.925 (0.889;0.949) |
| SEM | 1.18 | 0.88 |
| MDC | 3.28 | 2.45 |
| CV (%) | 38.7 | 20.1 |

n: Number; CV: Coefficient of Variation; ICC: Intraclass Correlation Coefficient; SEM: Standard Error of Measurement; MDC: Minimal Detectable Change.

## Discussion

This investigation is, to our knowledge, the first to examine both intra- and inter-examiner reliability of SWE in evaluating PM muscle stiffness in individuals with LBP. Among the key findings, test-retest reliability demonstrated to be high by both novice and experienced examiners, with all reliability indices yielding near-perfect ICCs (>0.9). Despite these strong ICC values, analysis of mean differences revealed that the measurements taken by the experienced examiner were more consistently reproducible. In contrast, although the novice examiner also achieved excellent ICCs, significant trial-to-trial variability was evident, suggesting reduced repeatability relative to the experienced examiner.

Regarding inter-examiner agreement, significant differences in mean scores were identified between examiners, despite good ICC values for both shear wave speed and Young's modulus. However, when the mean of two measurements was calculated, no significant differences were observed between examiners for Young's modulus. This approach improved agreement and accuracy, as evidenced by lower absolute error, SEM, and MDC values. Therefore, conducting a second measurement and calculating the mean average is justified for novel examiners for improving intra-examiner reliability. While experienced examiners obtained almost perfect ICCs obtained for a single measurement (>0.97) with no significant differences between trials, novice examiners have a wider improvement margin (as significant differences between trials were found). Therefore, even if the small improvement margin and the extra time required for multiple measurements is not worthy for experienced examiners, novice examiners may benefit from calculating a mean average of multiple trials.

The variability observed in both intra- and inter-examiner reliability when novice examiners were involved may be attributed to inconsistencies in transducer manipulation, including fluctuations in applied pressure, angulation or alignment. Additionally, differences in the interpretation of SWE images could have contributed to this variability, specially regarding precise delineation of the muscle boundaries [32], which may be hindered by potential histological alterations commonly associated with chronic pain [21,33–37].

Future studies should explore these hypotheses by involving novice and experienced examiners assessing healthy subjects. If reliability differences persist in healthy individuals, this would support the transducer handling hypothesis. Conversely, reduced variability in healthy subjects would point toward the impact of muscle histological changes in patients with LBP.

A comparison with Zhou et al. [23] (who reported ICCs of 0.89–0.92 using an examiner with 16 years of clinical experience and 3 years of SWE experience) suggests that experienced examiners are less affected by potential histological changes, supporting the importance of extensive training and experience in probe handling. The reliability observed in Zhou's study aligns with the results obtained here, regardless of whether participants had pain or were asymptomatic.

The comparison of stiffness scores from asymptomatic individuals reported by Zhou et al. [23] to those in this study should be interpreted with caution as different devices were used and there are no established reliability values for deep muscles. Nevertheless, this contrast revealed higher Young's modulus values in LBP patients (13.8 ± 3.7 kPa in asymptomatic individuals versus 15.79 ± 4.91 kPa in LBP patients). This difference may indicate that LBP, particularly through its chronic nature, contributes to increased muscle stiffness, possibly linked to the high prevalence of myofascial trigger points in the PM. These differences in stiffness between healthy and LBP populations should be highlighted considering that the mean BMI of the population in our study exceeded 25 points, indicating overweight or obesity, which could imply a greater presence of fatty infiltration in the analyzed muscles [38]. The presence of fatty infiltrates may reduce tissue stiffness and therefore be associated with a lower Young's modulus, indicating that, in subjects with normal and comparable BMIs, this contrast could be even more notable [39]. Further research is needed to explore whether SWE can effectively differentiate between LBP patients and asymptomatic individuals, the impact of BMI on SWE errors, and its correlation with clinical severity indicators of LBP.

To enhance measurement accuracy, for longitudinal or follow-up assessments, the participation of an experienced examiner is particularly important. Their lower MDC values allow for more accurate identification of true physiological changes in muscle stiffness, thereby decreasing the risk of mistaking measurement variability for actual clinical change. When posible, a single examiner is recommended to be responsible for both acquiring and interpreting SWE images in both clinical practice and research contexts. Variability in technique across different examiners can introduce inconsistencies, potentially leading to misinterpretation of changes in muscle stiffness. When the involvement of multiple examiners is unavoidable, averaging at least two measurements is advised, as this approach minimizes error and enhances measurement reliability, making it appropriate for both clinical and investigative use.

All these findings reinforce the critical role of examiner expertise and methodological rigor in obtaining reliable SWE measurements, particularly in clinical populations.

Despite the valuable results, this study has some limitations that should be acknowledged. The SWE-obtained parameters have not been compared with other clinical variables like the Thomas test, the range of motion in hip extension or the previous physical activity levels, factors that could enrich the clinical contribution of our research. In addition, the sample size is small and the design does not include a group of healthy subjects, which would allow to contrast the results of individuals with similar characteristics except for the presence or absence of LBP. Additionally, the fact that we only evaluated one portion of the PM at rest can restrict the generalizability of our results. Future studies could improve these aspects in their methodology to complement our findings and facilitate their extrapolation and integration in clinical settings.

## Conclusion

SWE test-retest reliability for evaluating PM muscle stiffness in individuals with chronic LBP was excellent for both novice and experienced examiners, with ICCs exceeding 0.9, providing highly consistent measurements when performed by the same assessor. In contrast, inter-examiner reliability was notably lower, reflecting considerable variability in SWE outcomes between examiners with differing levels of experience. The accuracy of SWE largely depends on consistency in the evaluator conducting the assessment. When multiple assessors must be involved, averaging multiple measurements is recommended to improve reliability and reduce the potential for measurement error.

## Supporting information

**S1 File. Raw data.**
(PDF)

## Author contributions

**Conceptualization:** Gabriel Rabanal-Rodríguez, Jorge Buffet-García, Juan Antonio Valera-Calero.

**Data curation:** Gabriel Rabanal-Rodríguez, Jorge Buffet-García, Juan Antonio Valera-Calero.

**Formal analysis:** Juan Antonio Valera-Calero.

**Funding acquisition:** Juan Antonio Valera-Calero.

**Investigation:** Gabriel Rabanal-Rodríguez, Javier Álvarez-González, Alberto Roldán-Ruiz, Jorge Buffet-García, Juan Antonio Valera-Calero.

**Methodology:** Gabriel Rabanal-Rodríguez, César Fernández-de-las-Peñas, Jorge Buffet-García, Juan Antonio Valera-Calero.

**Project administration:** Juan Antonio Valera-Calero.

**Resources:** Javier Álvarez-González, Alberto Roldán-Ruiz, Jorge Buffet-García, Juan Antonio Valera-Calero.

**Software:** Juan Antonio Valera-Calero.

**Supervision:** Jorge Buffet-García, Juan Antonio Valera-Calero.

**Validation:** César Fernández-de-las-Peñas, Juan Antonio Valera-Calero.

**Visualization:** César Fernández-de-las-Peñas, Juan Antonio Valera-Calero.

**Writing – original draft:** Gabriel Rabanal-Rodríguez, Juan Antonio Valera-Calero.

**Writing – review & editing:** Gabriel Rabanal-Rodríguez, César Fernández-de-las-Peñas, Jorge Buffet-García, Juan Antonio Valera-Calero.

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
