## [Decision Letter · Decision Letter 0]

23 Jun 2025

Dear Dr. Buffet-García,

Thank you for submitting your manuscript to PLOS ONE. After careful consideration, we feel that it has merit but does not fully meet PLOS ONE’s publication criteria as it currently stands. Therefore, we invite you to submit a revised version of the manuscript that addresses the points raised during the review process.

We look forward to receiving your revised manuscript.

Kind regards,

Masatoshi Nakamura, Ph.D.

Academic Editor

PLOS ONE

Additional Editor Comments (if provided):

Reviewers' comments:

Reviewer's Responses to Questions

**Comments to the Author**

1. Is the manuscript technically sound, and do the data support the conclusions?

Reviewer #1: Yes

Reviewer #2: Partly

2. Has the statistical analysis been performed appropriately and rigorously?

Reviewer #1: Yes

Reviewer #2: Yes

3. Have the authors made all data underlying the findings in their manuscript fully available?

Reviewer #1: Yes

Reviewer #2: Yes

4. Is the manuscript presented in an intelligible fashion and written in standard English?

Reviewer #1: Yes

Reviewer #2: Yes

Reviewer #1: Thank you for the opportunity to review this study. The present manuscript investigates the intra- and inter-rater reliability of shear wave elastography (SWE) measurements of the psoas major (PM) muscle. The study is well designed and clearly written, and the results suggest that SWE may be a reliable method for assessing PM stiffness in patients with low back pain (LBP). I appreciate the authors’ contribution to this area of musculoskeletal research. Nevertheless, I have several comments that I believe would strengthen the manuscript, particularly regarding the methodology.

Introduction:

-The Introduction is somewhat lengthy. The detailed description of the PM functional anatomy may be reduced or moved to the Discussion if necessary.

Methods:

-Was participants’ physical activity level controlled prior to the SWE assessment? For example, performing physical exercise or manual labor immediately before measurement could influence the mechanical properties of the PM. This potential confounder should be discussed

-Given that the PM is located relatively deep, was the depth of the region of interest (ROI) recorded during SWE acquisition? Depth can influence modulus values. If the depth varied across measurements or between participants, this should be considered in the interpretation.

-In each SWE measurement session, was only one SWE image acquired? If so, this should be stated clearly. Since SWE values tend to be quite sensitive to how transducer is manipulated, acquiring and averaging multiple scans would reduce measurement error and improve reliability. Please clarify.

-The classification thresholds for intraclass correlation coefficient (ICC) (e.g., moderate, good, excellent) should be explicitly defined, preferably with references.

-It would enhance the clinical impact of the study if SWE values were compared to physical examination findings associated with PM stiffness, such as the Thomas test or hip extension range of motion.　Furthermore, including healthy volunteers as a comparison group would have helped interpret whether elevated stiffness values are indeed associated with LBP.

Results:

-There seems to be a discrepancy in participant numbers. Table 1 reports 16 men and 31 women, which sums to 47, yet the total number of participants is stated as 52. Please confirm and correct this inconsistency.

Discussion

-Lines 338–344: The average BMI of both male and female participants exceeds 25, indicating that some may be classified as overweight or obese. Obesity can lead to intramuscular fatty infiltration, including in the PM. Could the relatively higher Young’s modulus observed in this study be attributed, at least in part, to fatty infiltration rather than LBP per se?

-Please add the limitations of this study.

Table

-As mentioned in the main text, please include p-values to support the statistical significance of the comparisons.

Reviewer #2: Thank you for the opportunity to review this manuscript aimed to evaluate the intra and inter-examiner reliability of shear wave elastography for calculating the PM stiffness in patients with LBP. While the approach is rigor and the findings are important, I would like to point out some methodological and interpretation concerns.

Major comments

1. While psoas major CSA changes and fatty infiltration were clearly described in L98-112, these changes did not necessarily align with the main aim, which was to evaluate the reliability of stiffness measurements. The introduction should reduce or eliminate these descriptions to focus on the need for a validated method to quantify muscle stiffness in LBP patients.

2. To further strengthen the rationale for this study, the authors should mention the prior reliability study in healthy subjects by Zhou et al. in the introduction. Following this, the authors should hypothesize why these findings of high reliability may not be generalizable to patients with LBP.

3. L181. The study analyzes muscles from both sides of the body. It is possible that data from the left and right sides of the same participant are not independent. I wonder if this lack of independence can cause the intraclass correlation coefficient (ICC) to increase. The potential influence of this non-independence on the statistical results should be discussed or addressed within the statistical model.

4. L254. While the authors assessed consistency within and between examiners using ICC(3,1) and ICC(3,2), they may overlook systematic error, which would be an issue in clinical settings. To provide a more comprehensive evaluation of reliability, I suggest adding an ICC(2,1), which incorporates both random and systematic error. Also, please calculate the SEM and MDC using the ICC(2,1) value.

Specific comments

Abstract

5. L66. The conclusion states, "Excellent test-retest reliability was observed for the measurement of PM muscle stiffness... by both experienced and novice evaluators." However, this statement should be revised to acknowledge the statistically significant difference between the two trials for the novice examiner.

Introduction

6. L95�”including” is a typo.

Methods

7. L191. What is the rationale behind including 52 participants, despite having confirmed that the minimum sample size was 29? Could using a larger sample size lead to an overestimation of reliability?

8. L232. Please add the region of interest size.

9. The manuscript does not explain how Young's modulus was calculated from the shear wave speed. To ensure reproducibility, the authors must provide the formula used and any assumptions made (e.g., tissue density).

10. Please add the information on imaging processing, such as smoothing an so on.

11. The authors have two values of each muscle in each examiner. Which data (trial 1 or 2?) do the author use to calculate ICC(3,1)?

Results

12. L264: The statement "Every image acquired during data collection was deemed valid" is concerning. Were there pre-defined criteria for image acceptability? The absence of any excluded images in 416 acquisitions seems unlikely and requires explanation.

13. L265. The manuscript has stated that the total number of participants is 52. However, 31 females and 16 males add up to 47. Please correct.

14. L267. Did you acquire two consecutive images without removing the probe, or did you reposition the probe before the second measurement?

15. L272. This section includes data for "Water volume" and “body composition”, but the method to measure these variables is not described in the Methods section.

16. Please consider adding the coefficient of variation alongside ICC to provide a more complete assessment of measurement variability, especially given the differing scales of shear wave velocity and Young’s modulus.

17. Bland-Altman plots are presented in the results, but this analysis is not described in the Methods section. Furthermore, the authors should analyze the presence or absence of systematic and proportional error using the plots and statistical tests.

Discussion

18. L338. In the Discussion, the authors compared their Young's modulus values with data from the study by Zhou et al. However, the two studies used different ultrasound systems (Canon vs. Aixplorer). Without established inter-machine reliability, particularly for deep muscles, this direct comparison is not valid. Therefore, I believe this comparison should be avoided.

19. The authors should acknowledge that evaluating only one portion of the PM at rest is a limitation that restricts the generalizability of the study.

Figure/Caption

20. Figure 1: The figure, particularly the left-hand schematic, appears very similar to a figure in the publication with PMID: 40150065. If this figure has been adapted or inspired by another source, that source must be appropriately cited. Also, in Figure1’s caption, “mayor” is a typo.

21. Figure 2: The caption for the Bland-Altman plots should explain what the solid red lines and the dashed lines represent. Additionally, the dashed lines appear curved in the figure; please confirm if this is an error or an intended feature of the plotting method.

22. Table2. In trial 2, the novice examiner measured a mean shear wave speed of 2.09 m/s and a Young’s modulus of 16.03 kPa, while the experienced examiner measured a higher shear wave speed of 2.15 m/s but a lower Young’s modulus of 15.9 kPa. Is this discrepancy due to different calculation methods, or is it a reporting error?

23. Table 3. I was not sure how the mean, error, and absolute error were calculated.

**Do you want your identity to be public for this peer review?** For information about this choice, including consent withdrawal, please see our Privacy Policy

Reviewer #1: No

Reviewer #2: No

---

## [Author Response · Author response to Decision Letter 1]

9 Jul 2025

Response to Reviewers

Shear Wave Elastography for measuring the elastic properties of the Psoas Major muscle: an intra- and inter-examiner reliability study

We would like to thank the Editor and Reviewers for their comments and suggestions for the improvement of our manuscript. We have carried out an in-depth revision to ensure that the document is adapted to the style requirements of the journal and to respond to the Reviewers' proposals. Changes are detailed below:

Academic Editor

Response: We have revised the formal aspects of the text and files to adapt them to the journal's requirements.

Response: Thank you. We confirm that our submission contains all required data.

Response: Thank you for your recommendation. We confirm that all data is reported in the manuscript. If any additional information is required, we will be pleased to provide it.

Response: We have deleted the ethics statement from the Declarations section and have left it only in the Methods section following your recommendations.

Reviewer #1

Thank you for the opportunity to review this study. The present manuscript investigates the intra- and inter-rater reliability of shear wave elastography (SWE) measurements of the psoas major (PM) muscle. The study is well designed and clearly written, and the results suggest that SWE may be a reliable method for assessing PM stiffness in patients with low back pain (LBP). I appreciate the authors’ contribution to this area of musculoskeletal research. Nevertheless, I have several comments that I believe would strengthen the manuscript, particularly regarding the methodology.

Response: Thank you for this positive feedback.

Introduction:

-The Introduction is somewhat lengthy. The detailed description of the PM functional anatomy may be reduced or moved to the Discussion if necessary.

Response: We appreciate your suggestion. We have reduced the detailed description of the PM anatomy to focus on its influence on LBP.

Methods:

-Was participants’ physical activity level controlled prior to the SWE assessment? For example, performing physical exercise or manual labor immediately before measurement could influence the mechanical properties of the PM. This potential confounder should be discussed.

Response: Despite being factors that could affect muscle properties, our study design did not take them into account as these factors are hard to control in the clinical practice. However, even if these factors were not controlled, the results we obtained further reinforce the real reliability of the procedure. Nevertheless, we have included this aspect in our limitations.

-Given that the PM is located relatively deep, was the depth of the region of interest (ROI) recorded during SWE acquisition? Depth can influence modulus values. If the depth varied across measurements or between participants, this should be considered in the interpretation.

Response: The depth of the region of interest varied between participants depending on their anthropometric characteristics, without setting a fixed reference value, as we have clarified in the Methods section. In our opinion, as discussed in the previous section and far from being a negative aspect, these possible variations reinforce the good reliability results that we obtained.

-In each SWE measurement session, was only one SWE image acquired? If so, this should be stated clearly. Since SWE values tend to be quite sensitive to how transducer is manipulated, acquiring and averaging multiple scans would reduce measurement error and improve reliability. Please clarify.

Response: Thank you. To avoid doubts, we have clarified that a single SWE image was acquired per muscle in each measurement session. Participants were asked to attend at four different times. The first two visits were in day one (in the morning with examiner A and in the afternoon with examiner B) and the second two visits were in day two (in the morning with examiner B and in the afternoon with examiner A).

Regarding the impact of averaging multiple scans, this is already discussed (declaring that a mean average calculation improved inter-examiner concordance compared to a single measurement, reaching no significant differences in Young’s modulus).

If you ask to calculate a mean average for improving intra-examiner reliability, we believe that investing time in conducting a second measurement and calculating the mean average is justified for novel examiners. Experienced examiners obtained almost perfect ICCs obtained for a single measurement (>0.97), with no significant differences between trials. Therefore, the small improvement margin and the extra time required for multiple measurements is not worthy. In contrast, novice examiners have more margin (as significant differences between trials were found) and they may benefit from calculating a mean average of multiple trials. We included this reflection in Discussion.

-The classification thresholds for intraclass correlation coefficient (ICC) (e.g., moderate, good, excellent) should be explicitly defined, preferably with references.

Response: Thank you for your appreciation, we have included the interpretation of the ICCs following the recommendations of Koo et al. (https://doi.org/10.1016/j.jcm.2016.02.012) in the Statistical analysis section.

-It would enhance the clinical impact of the study if SWE values were compared to physical examination findings associated with PM stiffness, such as the Thomas test or hip extension range of motion. Furthermore, including healthy volunteers as a comparison group would have helped interpret whether elevated stiffness values are indeed associated with LBP.

Response: Thank you very much for your suggestion. We totally agree and we are actually working on it, but this would require a totally different design (case-control study) and including both designs in a single article would increase significantly the length of the manuscript. While we recognize the clinical relevance of these studies, we believe that the first step should be confirm the reliability of the procedure to focus posteriorly on observational studies (correlation between muscle stiffness with clinical severity indicators, sensitivity and specificity to classify asymptomatic individuals or clinical populations, differences between cases and controls…)

We have added these aspects to the limitations and will take them into account for future research.

Results:

-There seems to be a discrepancy in participant numbers. Table 1 reports 16 men and 31 women, which sums to 47, yet the total number of participants is stated as 52. Please confirm and correct this inconsistency.

Response: Thank you for your comment, we apologize for the inconsistency. We have corrected the corresponding data on the total number of 47 subjects.

Discussion

-Lines 338–344: The average BMI of both male and female participants exceeds 25, indicating that some may be classified as overweight or obese. Obesity can lead to intramuscular fatty infiltration, including in the PM. Could the relatively higher Young’s modulus observed in this study be attributed, at least in part, to fatty infiltration rather than LBP per se?

Response: Thank you for your comment. We cannot confirm or discard this hypothesis as we did not conduct a correlation analysis. These analyses are planned after publishing the reliability results of the procedure (not only the association between SWE scores and BMI, but also the influence of BMI on SWE errors). For this purpose, a larger sample size and a wider range of BMIs would be needed to support the conclusions on enough statistical power.

We believe that this comment is a good opportunity to introduce the possible implication of BMI and fatty infiltration on SWE results. The possible increase of fatty infiltration in our sample could have reduced tissue stiffness, thus associating lower values of Young's modulus and making the differences between subjects with comparable BMIs even more significant.

-Please add the limitations of this study.

Response: Thank you. We have added this section at the end of the Discussion.

Table

-As mentioned in the main text, please include p-values to support the statistical significance of the comparisons.

Response: Thank you for your suggestions. We have added the p-values of the comparisons to the tables.

Reviewer #2

Thank you for the opportunity to review this manuscript aimed to evaluate the intra and inter-examiner reliability of shear wave elastography for calculating the PM stiffness in patients with LBP. While the approach is rigor and the findings are important, I would like to point out some methodological and interpretation concerns.

Response: Thank you for this positive feedback.

Major comments

1. While psoas major CSA changes and fatty infiltration were clearly described in L98-112, these changes did not necessarily align with the main aim, which was to evaluate the reliability of stiffness measurements. The introduction should reduce or eliminate these descriptions to focus on the need for a validated method to quantify muscle stiffness in LBP patients.

Response: Thank you for your recommendation. We have eliminated these descriptions to avoid information overload and to focus on the objective of the study.

2. To further strengthen the rationale for this study, the authors should mention the prior reliability study in healthy subjects by Zhou et al. in the introduction. Following this, the authors should hypothesize why these findings of high reliability may not be generalizable to patients with LBP.

Response: Thank you for your appreciation. We have mentioned the previous study by Zhou et al. in our introduction and hypothesized that their results may not be extrapolated to individuals with LBP due to their specific muscular characteristics.

3. L181. The study analyzes muscles from both sides of the body. It is possible that data from the left and right sides of the same participant are not independent. I wonder if this lack of independence can cause the intraclass correlation coefficient (ICC) to increase. The potential influence of this non-independence on the statistical results should be discussed or addressed within the statistical model.

Response: We agree with the reviewer that measurements taken from both sides of the same participant cannot be considered fully independent data points, and this is an important methodological consideration that has been now discussed in the manuscript. However, we believe that this decision does not artificially inflate the ICC values reported. In fact, the inclusion of bilateral data could theoretically lead to underestimation or overestimation of ICCs.

In our study, we took several steps to mitigate the potential impact of non-independence: 1) repositioning protocols for both the patient and the probe were strictly standardized across all trials; (2) the order of side evaluation was randomized; and (3) previous research (e.g., doi: 10.3390/bioengineering10080904) has shown that anthropometric and sociodemographic factors are not significantly associated with SWE measurement errors. Moreover, our sample displayed considerable heterogeneity in these variables, which reduces the likelihood that demographic similarity introduced any systematic bias. In fact, in populations with musculoskeletal pain, asymmetries are often more pronounced, which further supports the relevance of analyzing each side independently for reliability purposes. Based on these considerations, we believe that the reported ICCs are robust and not substantially influenced by any lack of independence between sides. Please check the section “Examiners” where we addressed this comment.

4. L254. While the authors assessed consistency within and between examiners using ICC(3,1) and ICC(3,2), they may overlook systematic error, which would be an issue in clinical settings. To provide a more comprehensive evaluation of reliability, I suggest adding an ICC(2,1), which incorporates both random and systematic error. Also, please calculate the SEM and MDC using the ICC(2,1) value.

Response: After carefully reviewing the guidelines provided by Koo and Li (2016), we chose to report ICC(3,1) and ICC(3,2) based on a two-way mixed-effects model for consistency, which is more appropriate when the same raters are fixed and of primary interest, as is the case in our study. ICC(2,1), based on absolute agreement in a random-effects model, is generally applied when the goal is to generalize the results to a larger population of raters, which is not the case in our design.

Moreover, absolute agreement is more suitable for categorical or rater-dependent decisions, where any dis

---

## [Decision Letter · Decision Letter 1]

27 Jul 2025

Shear Wave Elastography for measuring the elastic properties of the Psoas Major muscle: an intra- and inter-examiner reliability study

PONE-D-25-25182R1

Dear Dr. Buffet-García,

We’re pleased to inform you that your manuscript has been judged scientifically suitable for publication and will be formally accepted for publication once it meets all outstanding technical requirements.

Kind regards,

Masatoshi Nakamura, Ph.D.

Academic Editor

PLOS ONE

Additional Editor Comments (optional):

Reviewers' comments:

Reviewer's Responses to Questions

**Comments to the Author**

Reviewer #1: All comments have been addressed

Reviewer #2: All comments have been addressed

2. Is the manuscript technically sound, and do the data support the conclusions?

Reviewer #1: Yes

Reviewer #2: Yes

3. Has the statistical analysis been performed appropriately and rigorously?

Reviewer #1: Yes

Reviewer #2: Yes

4. Have the authors made all data underlying the findings in their manuscript fully available?

Reviewer #1: Yes

Reviewer #2: Yes

5. Is the manuscript presented in an intelligible fashion and written in standard English?

Reviewer #1: Yes

Reviewer #2: Yes

Reviewer #1: The author has sufficiently addressed all of my comments. The revised manuscript has been improved accordingly and is now suitable for publication.

Reviewer #2: Thank you for addressing all my comments. I feel you have responded carefully to the concerns I raised. I have just one point, which may be a difference in perspective: while I agree with the interpretation of ICC(3,1) and ICC(2,1), I believe that since the SEM and MDC are also derived from ICC(3,1), they fail to account for the influence of systematic error.

**Do you want your identity to be public for this peer review?** For information about this choice, including consent withdrawal, please see our Privacy Policy

Reviewer #1: No

Reviewer #2: No

---

## [Editor Report · Acceptance letter]

PONE-D-25-25182R1

PLOS ONE

Dear Dr. Buffet-García,

I'm pleased to inform you that your manuscript has been deemed suitable for publication in PLOS ONE. Congratulations! Your manuscript is now being handed over to our production team.

Kind regards,

on behalf of

Dr. Masatoshi Nakamura

Academic Editor

PLOS ONE